# Deep mining of the Sequence Read Archive reveals major genetic innovations in coronaviruses and other nidoviruses of aquatic vertebrates

Chris Lauber[1,2]�*, Xiaoyu Zhang[1]�, Josef Vaas[3,4], Franziska Klingler[3,4], Pascal Mutz[3¤], Arseny Dubin[5], Thomas Pietschmann[1,2], Olivia Roth[5], Benjamin W. Neuman[6], Alexander E. Gorbalenya[7,8]*, Ralf Bartenschlager[3,4], Stefan Seitz[3,4]*

1 Institute for Experimental Virology, TWINCORE Centre for Experimental and Clinical Infection Research, a joint venture between the Hannover Medical School (MHH) and the Helmholtz Centre for Infection Research (HZI), Hannover, Germany, 2 Cluster of Excellence 2155 RESIST, Hannover, Germany, 3 Division of Virus-Associated Carcinogenesis (F170), German Cancer Research Center (DKFZ), Heidelberg, Germany, 4 Heidelberg University, Medical Faculty Heidelberg, Department of Infectious Diseases, Molecular Virology, Center for Integrative Infectious Disease Research, Heidelberg, Germany, 5 Marine Evolutionary Biology, Zoological Institute, Kiel University, Kiel, Germany, 6 Department of Biology and Texas A&M Global Health Research Complex, Texas A&M University, College Station, Texas, United States, 7 Leiden University Center of Infectious Diseases, Leiden University Medical Center, Leiden, The Netherlands, 8 Faculty of Bioengineering and Bioinformatics and Belozersky Institute of Physico-Chemical Biology, Lomonosov Moscow State University, Moscow, Russia

९ These authors contributed equally to this work.
¤ Current address: National Center for Biotechnology Information, National Library of Medicine, Bethesda, Maryland, United States
* chris.lauber@twincore.de (CL); a.e.gorbalenya@lumc.nl (AEG); s.seitz@dkfz-heidelberg.de (SS)

**Data Availability Statement:** We have uploaded a list of all analyzed SRA sequencing experiments, viral contig sequences generated in this study

## Abstract

Virus discovery by genomics and metagenomics empowered studies of viromes, facilitated characterization of pathogen epidemiology, and redefined our understanding of the natural genetic diversity of viruses with profound functional and structural implications. Here we employed a data-driven virus discovery approach that directly queries unprocessed sequencing data in a highly parallelized way and involves a targeted viral genome assembly strategy in a wide range of sequence similarity. By screening more than 269,000 datasets of numerous authors from the Sequence Read Archive and using two metrics that quantitatively assess assembly quality, we discovered 40 nidoviruses from six virus families whose members infect vertebrate hosts. They form 13 and 32 putative viral subfamilies and genera, respectively, and include 11 coronaviruses with bisegmented genomes from fishes and amphibians, a giant 36.1 kilobase coronavirus genome with a duplicated spike glycoprotein (S) gene, 11 tobaniviruses and 17 additional corona-, arteri-, cremega-, nanhypo- and nangoshaviruses. Genome segmentation emerged in a single evolutionary event in the monophyletic lineage encompassing the subfamily *Pitovirinae*. We recovered the bisegmented genome sequences of two coronaviruses from RNA samples of 69 infected fishes and validated the presence of poly(A) tails at both segments using 3'RACE PCR and subsequent Sanger sequencing. We report a genetic linkage between accessory and structural proteins whose phylogenetic relationships and evolutionary distances are incongruent with the

(Fasta format), sequence alignments (Fasta), variant calling files and phylogenetic trees (Newick) to FigShare: DOI 10.6084/m9.figshare.14874318. The Virushunter and Virusgatherer tools are available on github: https://github.com/lauberlab/VirusHunterGatherer.

**Funding:** CL and TP are supported by the Deutsche Forschungsgemeinschaft (DFG, German Research Foundation) under Germany's Excellence Strategy - EXC 2155 - project number 390874280. CL, RB and SS acknowledge support of the Project "Virological and immunological determinants of COVID-19 pathogenesis – lessons to get prepared for future pandemics (KA1-Co-02 "CoViPa")", a grant from the Helmholtz Association's Initiative and Network Fund. The coronavirus research in the lab of TP is supported by the Niedersächsisches Ministerium für Wissenschaft und Kultur (Ministry for Science and Culture of Lower Saxony) (grant 14-76103-184 CORONA-13/20). BWN acknowledges funding from the Texas A&M-Grants program OR is supported by the European Research Council (ERC) under the European Union's Horizon research and innovation program (MALEPREG: eu-repo/grantAgreement/EC/H2020/855659) and the German Research Foundation (RO-4628/9-1; RO4628/3-2; RO 4628/3-3). The funders had no role in study design, data collection and analysis, decision to publish, or preparation of the manuscript.

phylogeny of replicase proteins. We rationalize these observations in a model of inter-family S recombination involving at least five ancestral corona- and tobaniviruses of aquatic hosts. In support of this model, we describe an individual fish co-infected with members from the families *Coronaviridae* and *Tobaniviridae*. Our results expand the scale of the known extraordinary evolutionary plasticity in nidoviral genome architecture and call for revisiting fundamentals of genome expression, virus particle biology, host range and ecology of verte-brate nidoviruses.

## Author summary

Research in virology is primarily motivated by human pathogens, such as SARS-CoV-2 in the case of the family *Coronaviridae* in the order *Nidovirales*. Studies of these and few model viruses describe virus-host interactions on the molecular level and are essential for developing virus control measures, but they must accommodate a vast range of viral natu-ral diversity to allow generalizations. Here, we redefine our understanding of the genetic and genomic diversity in corona- and other nidoviruses of poorly sampled hosts. We mine more than 269,000 publicly accessible raw sequencing datasets for the presence of viral sequences using high-performance computing and discover 40 nidoviruses including 13 coronaviruses from a wide range of vertebrates. Some of the novel viruses from aquatic hosts have extraordinary features such as segmented genomes and recombinant genes coding for structural proteins. Our study suggests that gene exchange between diverse nidovirus species from different virus families might be more frequent than previously thought and can result in abrupt genomic innovations that in turn might facilitate host jumps even across vertebrate class borders. The growing list of newly discovered (corona) viruses enables an evolutionary perspective across virus divergency scales in different hosts on the wet lab-acquired knowledge about few viruses.

## Introduction

Nidoviruses form 14 virus families in the order *Nidovirales* for which the International Com-mittee on Taxonomy of Viruses (ICTV) currently recognizes 48 genera and 130 species in total [1–3]. The members of eight nidovirus families (*Arteriviridae*, *Coronaviridae*, *Cremega-viridae*, *Gresnaviridae*, *Nangoshaviridae*, *Nanhypoviridae*, *Olifoviridae*, *Tobaniviridae*) have vertebrate hosts while those of the remaining six families (*Abyssoviridae*, *Euroniviridae*, *Medio-niviridae*, *Mesoniviridae*, *Monidoviridae*, *Roniviridae*) infect invertebrates. The nidovirus fam-ily *Coronaviridae* has attracted unparalleled scientific and public attention due to the emergence of the human pathogens severe acute respiratory syndrome coronavirus (SARS--CoV) in 2002, Middle East respiratory syndrome coronavirus (MERS-CoV) in 2012 and the SARS-CoV-2 in 2019 [4–7], although studies on coronaviruses have fueled research in virology and beyond for decades [8].

A hallmark of corona- and other nidoviruses is the exceptionally large size range of their single-segment genomes that include at the upper end the 35.9 kb genome of Aplysia abysso-virus 1 (AAbV) [9] and the largest known RNA virus genome of 41.1 kb from the planarian secretory cell nidovirus (PSCNV) [10]. Most nidovirus genomes have the canonical architec-ture (from 5' to 3'): 5' untranslated region (5'UTR), open reading frame (ORF) 1a, ORF1b, 3'-proximal ORFs (3'ORFs) and 3'UTR. Products encoded in ORF1a/b are generated by

translation of the genomic RNA, comprising a -1 ribosomal frameshift in the region of the ORF1a/b overlap [11]. The 3'ORFs are expressed via subgenomic RNAs whose numbers vary between nidovirus species [12,13].

Comparative genomics played a pivotal role in advancing our understanding of coronaviruses and other nidoviruses by assigning putative functions to many nidovirus proteins, which were subsequently confirmed and elaborated in experimental studies [14–19]. All nidoviruses express a conserved array of five protein domains in ORF1a/ORF1b that control genome expression and replication. These include i) the 3C-like main protease (3CLpro or Mpro) flanked by highly variable but ubiquitous transmembrane domains, ii) the nidovirus RNA-dependent RNA polymerase (RdRp)-Associated Nucleotidyltransferase (NiRAN), iii) the RdRp, iv) a zinc-binding domain (ZBD) and v) a superfamily 1 helicase (HEL1). NiRAN and ZBD have no known virus homologs outside the nidoviruses and are therefore considered to be genetic markers of nidoviruses [15,20]. Nidoviruses with genomes exceeding 20 kb additionally encode an exoribonuclease (ExoN) with proofreading activity that has been linked to genome expansion by improving the otherwise low fidelity of the RdRp [20–22].

Coronaviruses express four structural proteins from their 3'ORFs encoded in the order: spike glycoprotein (S), envelope protein (E), matrix protein (M) and nucleocapsid phosphoprotein (N); this genome region may also encode non-essential accessory proteins [23,24]. The C-terminal half of S (S2) is well conserved within the family *Coronaviridae* and includes determinants of lipid association and infectivity; this conservation partly extends also to the sister family *Tobaniviridae* (formerly known as subfamily *Torovirinae*, family *Coronaviridae*) [25]. Otherwise, there is little to no similarity reported at the sequence level between coronaviral structural proteins and those of other nidoviruses.

There is accumulating evidence for homologous recombination involving various genomic regions including the S ORF within coronavirus species and between coronaviruses of closely related species [26–29]. We are not aware of comparable evidence for homologous recombination between members of different nidovirus families, although such studies may be complicated by the limited inter-family conservation. On the other hand, a pivotal role of heterologous recombination in the generation of nidovirus diversity is documented. It is evident in the duplication of protein domains and the restricted phyletic distribution of many conserved proteins to some nidovirus lineages and their phylogenetic links to non-nidovirus homologs [30]. In this respect, nidoviruses are no different than other RNA and DNA viruses [31–34]. The above characterization is dominated by studies involving comparative genomics which define our understanding about the type and scale of recombination. Quantifying incongruences of phylogenetic trees for different genome regions is a main approach to the characterization of homologous recombination in RNA viruses, including nidoviruses [27,29,35,36].

The presence of characteristic nidoviral protein domains, ZBD and NiRAN, and phylogenetic clustering using RdRp and HEL1 allow reliable identification of nidoviruses by comparative genomics. Following the emerging trend in virology, recently described nidoviruses have been discovered by bioinformatics analysis of next or third generation sequencing data from meta-genomic and -transcriptomics studies of diverse specimens [9,10,37–50]. These datasets are composed of overlapping sequence fragments of variable lengths (so-called reads) and various origins that can be assembled into contigs, some of which may represent full-length or partial viral genomes. Discrimination of viral from non-viral contigs is typically achieved by sequence-based comparisons involving known reference organisms, including viruses. Depending on the sensitivity of the method and the degree of divergence of the sequences in a sample, a fraction of sequences usually remains unclassified; this sequence space is often called 'dark matter' and may include undescribed highly divergent viruses [51]. Both assembled and

unprocessed sequencing data are deposited in public databases, making them available for (re-)analysis by the scientific community. Examples include the Transcriptome Shotgun Assembly (TSA) database, the Whole Genome Shotgun (WGS) database and the Sequence Read Archive (SRA) whose sizes grow with a non-linear rate. The latter stores unprocessed, primary sequencing data along with often detailed metadata annotation, and it has been demonstrated that the SRA and similar data repositories are a rich source of hitherto unknown novel viral sequences [33,52–55].

Here we introduce an original, highly parallelized computing workflow that has a sequence homology search with advanced sensitivity at its core and implements a targeted assembly approach to reconstruct full-length viral genome sequences. By applying this approach to more than 269,000 SRA datasets we reconstructed genome sequences of numerous vertebrate nidoviruses. A subset thereof are prototype members of 18 tentative novel genera of nidoviruses, including novel coronavirus genera as well as five tentative novel nidovirus subfamilies. The newly discovered viruses include 11 coronaviruses with bisegmented genomes that form a monophyletic lineage and are members of subfamily *Pitovirinae*, family *Coronaviridae* infecting aquatic hosts. We recover the sequences of both segments from newly generated RNA samples of 69 infected fishes. Moreover, we describe a new coronavirus with a giant genome of 36.1 kb that encodes two genes with significant sequence similarity to the S gene of other corona- and tobaniviruses. The identification of leader and body transcription regulatory sequences (TRSs) provide evidence for the production of subgenomic RNAs in two of the discovered nidoviruses. Our comparative genomics and phylogenetic analyses suggest a possible swapping of structural proteins between ancestors of subsets of corona- and tobaniviruses.

## Results

### A virus discovery approach targeting low sequence similarities in unprocessed SRA data

Our three-stage computational approach involves the sensitive sequence similarity-based detection of viral sequence reads in a raw sequence dataset followed by the assembly of full-length viral genome sequence(s) and virus assignment. To achieve this with reasonable speed, we queried the *in silico* translated primary sequencing data (raw reads) in a highly parallelized fashion using profile Hidden Markov Models (pHMMs) of proteins characteristic of a virus group, often including the RdRp, and a targeted viral genome assembly method. At the first stage, called Virushunter, we identified most conserved sequences of viruses that may belong to a group of interest. We aimed for ensuring detection of divergent viral sequences with sequence identity to viral reference proteins well below 35% in the 'twilight zone' of protein sequence similarity [56]. The identified hits served as seeds at the next stage, called Virusgatherer. These seeds were gradually extended with overlapping reads to assemble a genomic sequence or its segment as complete as technically feasible, which was one of the main objectives of this study. At the third stage, Virusassignment, the assembled sequence was assigned to the group of interest or another related virus group. We exclusively utilized non-commercial high performance computing infrastructure that is free of charge for scientific purposes.

With the aim to assure reliability of the viral genome assemblies and to enable future comparisons of assembly quality between different studies, we developed two novel metrics that quantitatively assess the per-base and contig-wide accuracy of the genome sequences (see Materials and Methods for details). These metrics are applicable across datasets and we foresee that this or similar approaches could be adopted as community standards. We considered to quantify genome completeness, defined here as coverage of the complete protein coding part of the genome, the complete 3'UTR including a poly(A) tail and a complete or partial 5'UTR.

However, we found it unrealistic, because this metric would require comparisons with closely related reference sequences that are rarely available, in particular for novel divergent viruses, which dominated our dataset.

To assess contig-wide sequence assembly accuracy, we computed the **MI**nimal **Co**verage Depth (MICO) of a contig. First, we determined the position(s) of the contig to which the lowest number of sequencing reads align and then declaring this number of reads to be the contig mico (Fig 1A). To deduce MICO values, we then mapped the mico to deciles of the mico distribution that we computed for a reference assembly set consisting of 2350 RNA virus sequences, which were assembled from vertebrate and invertebrate datasets by others and verified by Sanger sequencing [37,57]. This resulted in possible MICO values for our nidovirus contigs in the range of 1 to 10, with MICO = 1 assigned to contigs with mico values in the lowest 10% of mico values of the reference set, while MICO = 10 is given to contigs in the highest 10% of reference mico values. To assess the per-base accuracy of our contigs, we computed the **Me**an **A**lignment **S**core (MEAS) of a contig by calculating the average alignment score across all sequencing reads overlapping with a sequence position and then averaging this value across all sequence positions (meas) (Fig 1A). We mapped the meas values to deciles of a reference set distribution to derive MEAS, as we did for MICO.

The MICO and MEAS values are readily interpretable and comparable across datasets as a particular assembled sequence is assigned to one of ten possible quality categories defined by a reference population of assemblies. This reference population should be formed by assemblies of known and acceptable quality, and it could be continuously expanded in the future, further improving this quality assessment method.

## Discovery of novel vertebrate nidoviruses

With the aim to systematically screen unprocessed, primary sequencing data from the SRA databank we built on two earlier pilot studies [33,52] and analyzed 269,184 transcriptomic datasets. We included in our screen sequencing runs from vertebrates excluding those from highly over-represented model organisms like zebrafish, mouse and rat, as well as human (as of February 2022 for fishes and September 2020 for other vertebrates). For control purposes, we included several SRA transcriptome datasets of *Aplysia californica*, *Schmidtea mediterranea* and *Microhyla fissipes* from which three divergent nidoviruses had been discovered recently [9,10]. The total selection amounted to 428.6 terabyte of (compressed) data.

We used our newly developed approach to scan the vertebrate SRA data (Virushunter stage) and obtained 1,924 significant hits (E-value $< 1 \times 10^{-4}$) in 0.6% of the analyzed sets that comprised a large variety of host taxa (Fig 2). We then conducted a targeted virus genome assembly for these SRA datasets (Virusgatherer stage) by starting with a seed formed by the respective sequencing reads identified in the first Virushunter stage and iteratively extending the contig sequence using additional reads that (partially) align to the contigs ends, until no further matching reads were found. The resulting contigs were subsequently filtered to remove any remaining non-viral sequences (blastx against non-viral portion of RefSeq-Protein with E-value cut-off of $1 \times 10^{-4}$) and to retain sequences of at least 1000 nucleotides in length. Using a profile search specific for nidoviruses with nidovirus-wide NiRAN and RdRp pHMMs as query, we selected those contigs that showed highest sequence similarity; we considered the selected contigs as belonging to nidoviruses (Virusassignment stage; see Materials and Methods for further details). The majority of assembled viral sequences were found in samples from host orders Artiodactyla and Primates and often matched Porcine reproductive and respiratory syndrome virus (PRRSV) 1, PRRSV-2, MERS-CoV or SARS-CoV that were used in experimental infections of the respective laboratory animals (Fig 2). Likewise, we also reassembled

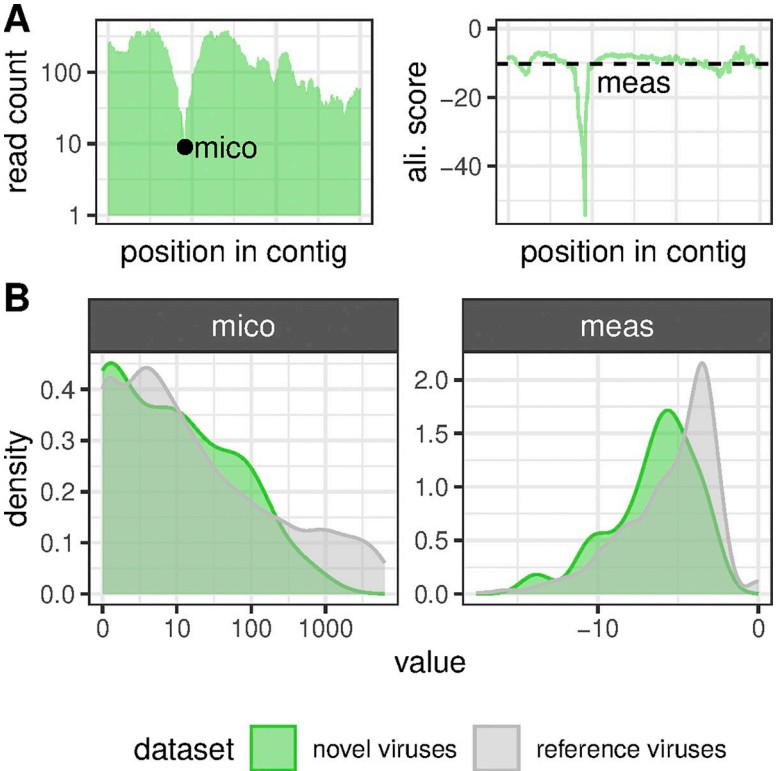

**Fig 1. Assembly quality assessment.** (A) Toy example visualizing how meas (left) and mico (right) assembly quality metrics are calculated. Alignment scores used for meas were calculated using Bowtie2 and have a maximum value of zero corresponding to reads aligning full-length without mismatches. (B) Distribution of meas and mico values obtained for the nidoviral sequences discovered and assembled in this study (green) and for 2350 reference RNA virus sequences (gray) [37,57]. Both x-axes are in $log_{10}$ scale.

genomes of two recently described, divergent nidoviruses with giant genome sizes from a sea slug (*Aplysia californica*) and a flatworm (*Schmidtea mediterranea*) as well as a novel corona-like virus from a frog (*Microhyla fissipes*) identified in our screen [9,10]. These confirmatory results could be considered a positive control of the novel viruses discovered in unrelated experiments from a wide range of hosts; they further validate our approach.

Focusing our subsequent analyses on novel vertebrate nidoviruses related to members of the families *Coronaviridae*, *Tobaniviridae*, *Arteriviridae*, *Cremegaviridae*, *Gresnaviridae*, *Olifoviridae*, *Nanhypoviridae* and *Nanghoshaviridae*, we recognized novel vertebrate nidoviruses as belonging to a single population entity (putative virus species) if the respective genomes showed >90% nucleotide sequence identity. To account for the observation that a particular virus species could be identified in multiple SRA datasets, we merged the respective sequences to reduce the associated sequence redundancy and generated variant calling files for these cases (see Materials and Methods for details). If a viral contig clustered with a known reference virus under the operational virus species threshold it was assigned to be a variant of that species, otherwise it was designated to be the prototype of a novel species. This strategy resulted in the delineation of 40 tentative species of nidoviruses of which 21 were novel. The viral sequences were derived from various organs/tissues of putative vertebrate hosts (S1 Table). The 19 known nidoviruses putatively infecting vertebrates were independently discovered in another study using a different computational data mining pipeline [53], further supporting the validity of our approach and findings.

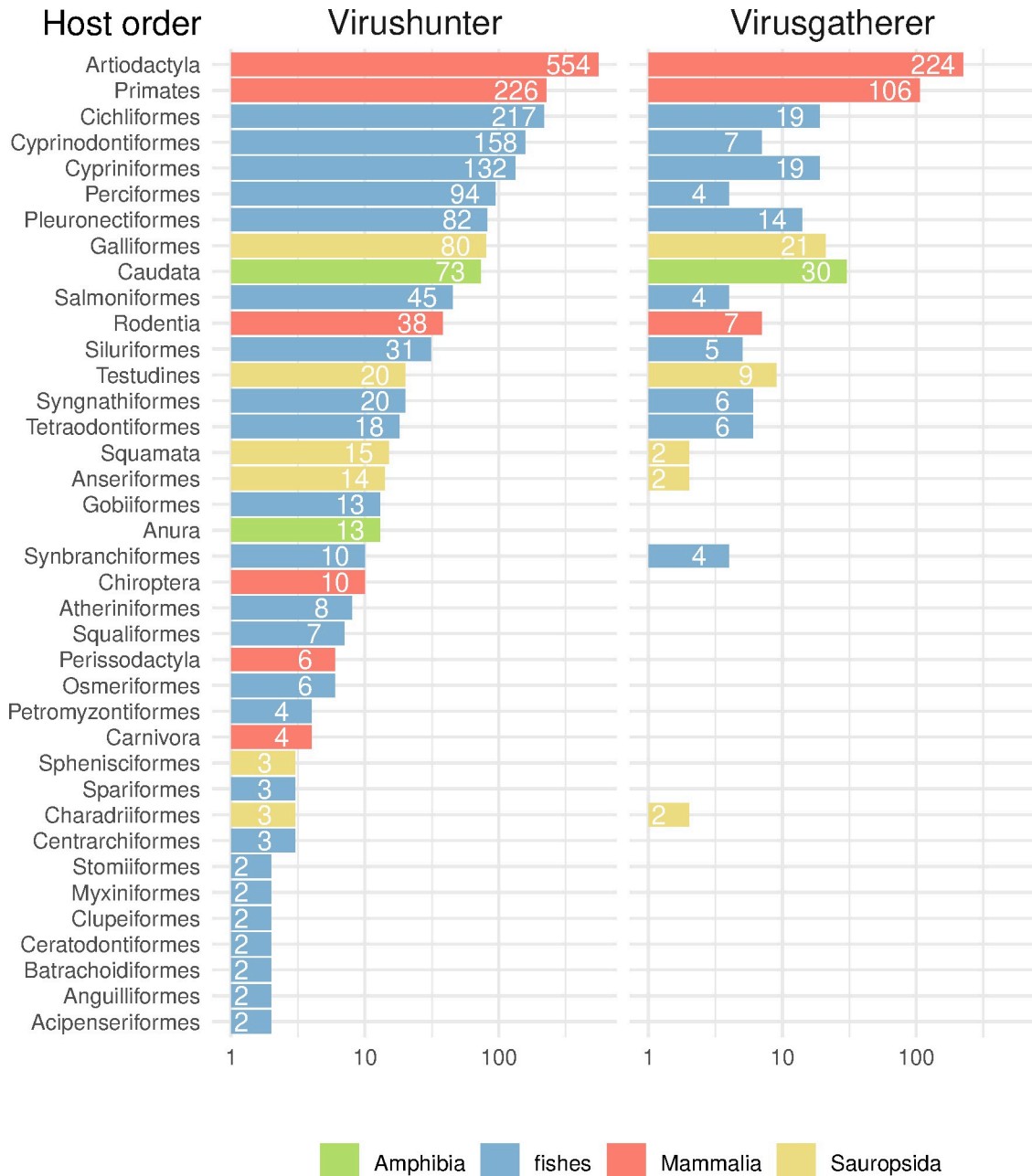

**Fig 2. Virus discovery in the SRA.** All numbers in the different panels correspond to counts of SRA runs. (Left) Results of a profile hidden Markov model (pHMM) based sequence homology search in the raw read data (Virushunter). Significant hits (at least one sequencing read with E-value $< 1x10^{-4}$) against one of three nidovirus pHMMs (see Materials and Methods for details) are shown if the corresponding sequences did not give better hits against other RNA viruses or against host sequences. Hits are grouped by order of the putative vertebrate host according to the annotation of the sequencing projects. Note that a detected sequence may not necessarily be from a member of the order *Nidovirales* but might also be from a virus of a related taxon for which no reference sequence was available by the time of analysis. (Right) Remaining hits after targeted viral genome assembly (Virusgatherer). Only contigs of at least 1000 nt in length were considered, and those with significant hits (covering at least 500 nt with E-value $< 1x10^{-4}$) against nidoviruses were kept. Bars are colored according to four major groups of the putative hosts (see common legend at the bottom-right).

When assessing viral genome assembly quality, we found that the MICO values of the novel vertebrate nidoviruses discovered in this study are in the same range as those of the reference set (Fig 1B), although slightly smaller on average (Wilcoxon rank-sum test, P = 0.020). The latter is expected as the reference sequences were from dedicated virus discovery projects favoring virus-enriched samples while our nidovirus sequences were not. At the level of an individual genome sequence, we observed almost the entire spectrum of MICO values for our novel nidoviruses (S1 Fig). Low MICO values concerned several partial fish coronavirus genomes with missing sequence. We estimated position and length of the missing pieces via comparison with the genome sequence of the most closely related virus known so far. We obtained reasonable to good coverage depth for the remaining viral contigs without internal missing sequence (S2 and S3 Figs). We again observed slightly lower MEAS values for the novel nidovirus contigs compared to the sequences of the reference set (Wilcoxon rank-sum test, P = 0.004) and a large spectrum of MEAS values (Figs 1B and S1).

## Novel corona- and tobaniviruses

Having described our computational approach and its accuracy, in the following we report the nidoviruses and their genomic properties that we discovered in our analysis. According to a RdRp+ZBD+HEL1-based phylogeny reconstruction and a genetics-based classification analysis by DEmARC [58], the 40 discovered putative virus species (species-like operational taxonomic units, sOTUs) can be grouped into 6 family-like operational taxonomic units (fOTUs), 13 subfamily-like OTUs (sfOTUs) and 32 genus-like OTUs (gOTUs) of which 5 sfOTUs and 18 gOTUs are novel (Fig 3A and S2 Table). All viruses with complete or nearly complete genome sequences encode a conserved array of nidovirus enzymes, 3CLpro-NiRAN-RdRp-ZBD-HEL1 and, additionally, a conserved O-Methyltransferase (OMT) domain was detected at the expected C-terminal position of ORF1b or its equivalent in viruses with large genomes (Figs 4 and S4) [59].

Thirteen of the discovered viruses clustered with known viruses within the family *Coronaviridae*: 12 detected in experiments with fish and one in the axolotl amphibian (*Ambystoma mexicanum*). These viruses comprise twelve new tentative genera outside the two established subfamilies *Orthocoronavirinae* and *Letovirinae*, according to our genetics-based classification. One of the new fish coronaviruses from *Hypomesus transpacificus* (HTCV) had a genome size of over 36 kb, making it, to the best of our knowledge, the largest RNA virus with monopartite genome infecting vertebrates (Fig 4). The genome seems to be coding-complete and ends with a poly(A) tail indicative of the genuine 3'-terminal end. The 36kb fish virus genome encodes two consecutive ORFs immediately downstream of ORF1b that both show significant sequence similarity to coronavirus spike proteins (Fig 4) and which share around 21% local protein sequence identity when compared to each other.

Eleven of the discovered coronaviruses have bisegmented genomes in which the first segment has coding regions for ORF1a and ORF1b, but no other ORFs, while the second segment encodes the structural proteins spike, matrix, and nucleocapsid as well as accessory proteins; they form an RdRp+ZBD+HEL1-based monophyletic cluster comprising subfamily *Pitovirinae* (Fig 4). Genome bisegmentation in these viruses was supported by the presence of a poly(A) tail at the 3'-end of many of the analyzed segments (Fig 4; see also below). Additionally, we did not identify any sequencing read pairs for which one mate aligns to the 3'-end of segment 1 and the other to the 5'-end of segment 2 in the order that would be expected if bipartition of the genome sequence was fortuitously due to sequencing or assembly artifacts (see Materials and Methods for details). The phylogenetic cluster of these bisegmented viruses includes an already described coronavirus (Pacific salmon nidovirus) from a fish species (*Oncorhynchus*

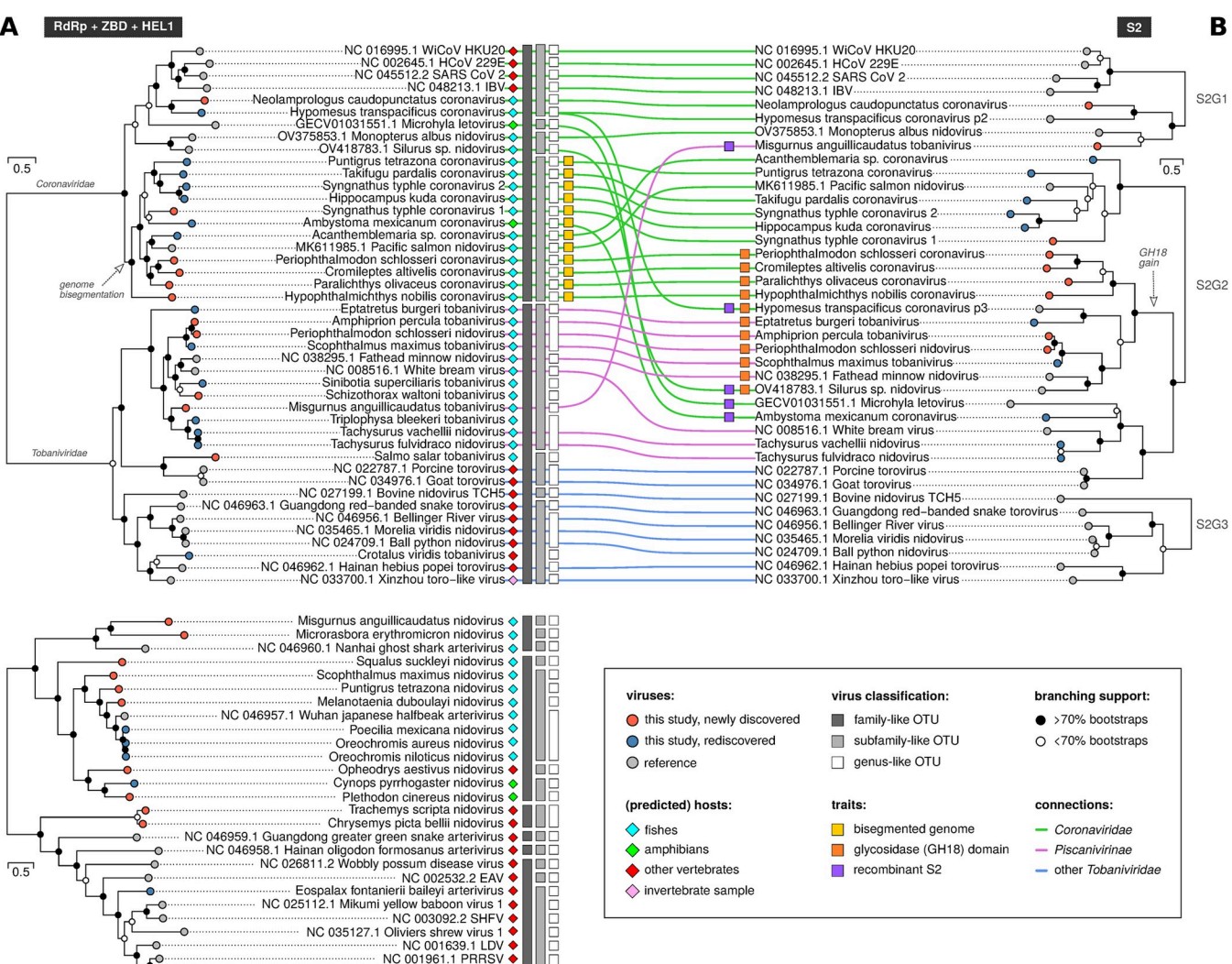

**Fig 3. Maximum likelihood phylogenies of non-structural and structural proteins and tanglegram of vertebrate nidoviruses.** The trees are based on protein alignments from which poorly conserved positions were manually removed. The phylogenies of non-structural proteins involving *Coronaviridae* and *Tobaniviridae* members (top) and *Nangoshaviridae*, *Nanhypoviridae*, *Gresnaviridae*, *Olifoviridae* and *Arteriviridae* members (bottom) are based on a concatenated alignment of RdRp, ZBD and HEL1 (A). The S protein phylogenies involving *Coronaviridae* and *Tobaniviridae* members (B) are based on conserved regions of the S2 part of the spike protein in coronaviruses or the homologous part in tobaniviruses. Two separate trees for A and three separate trees for B were constructed (see Materials and Methods for details). The branch lengths are in units of aa substitutions per site; scale bars are shown. White and black circles at internal nodes indicate branching support. Tips corresponding to reference viruses are shown as gray circles and those constituting lineages rediscovered or newly discovered from SRA data as blue and red circles, respectively. Family-like, subfamily-like and genus-like OTUs derived from a genetics-based classification using DEmARC are shown using dark gray, light gray and white rectangles, respectively; known or predicted host types are indicated by colored diamonds next to the virus names; viruses with bisegmented genomes, inferred recombinant S2 and those expressing a putative glycosidase domain are highlighted by colored squares (see legend at the bottom-right). Possible additional recombinant S2 cases are discussed in the text.

*tshawytscha*) [47]. It was originally annotated as monopartite in GenBank (accession MK611985), but does employ a bisegmented genome as well, according to our analysis.

This viral genome shows a unique insertion of a macrodomain locus in ORF1b between endonuclease (EndoU) and OMT that reside in nsp15 and nsp16 in experimentally characterized coronaviruses (Fig 4). We identified signal peptide cleavage sites for all *in silico* translated spike protein sequences of the unsegmented and bisegmented coronavirus genomes, with predicted signal peptide sizes in the range of 15 to 41 amino acids. Strikingly, for each coronavirus with bisegmented genome we observed an excess of sequencing reads mapping to segment 2

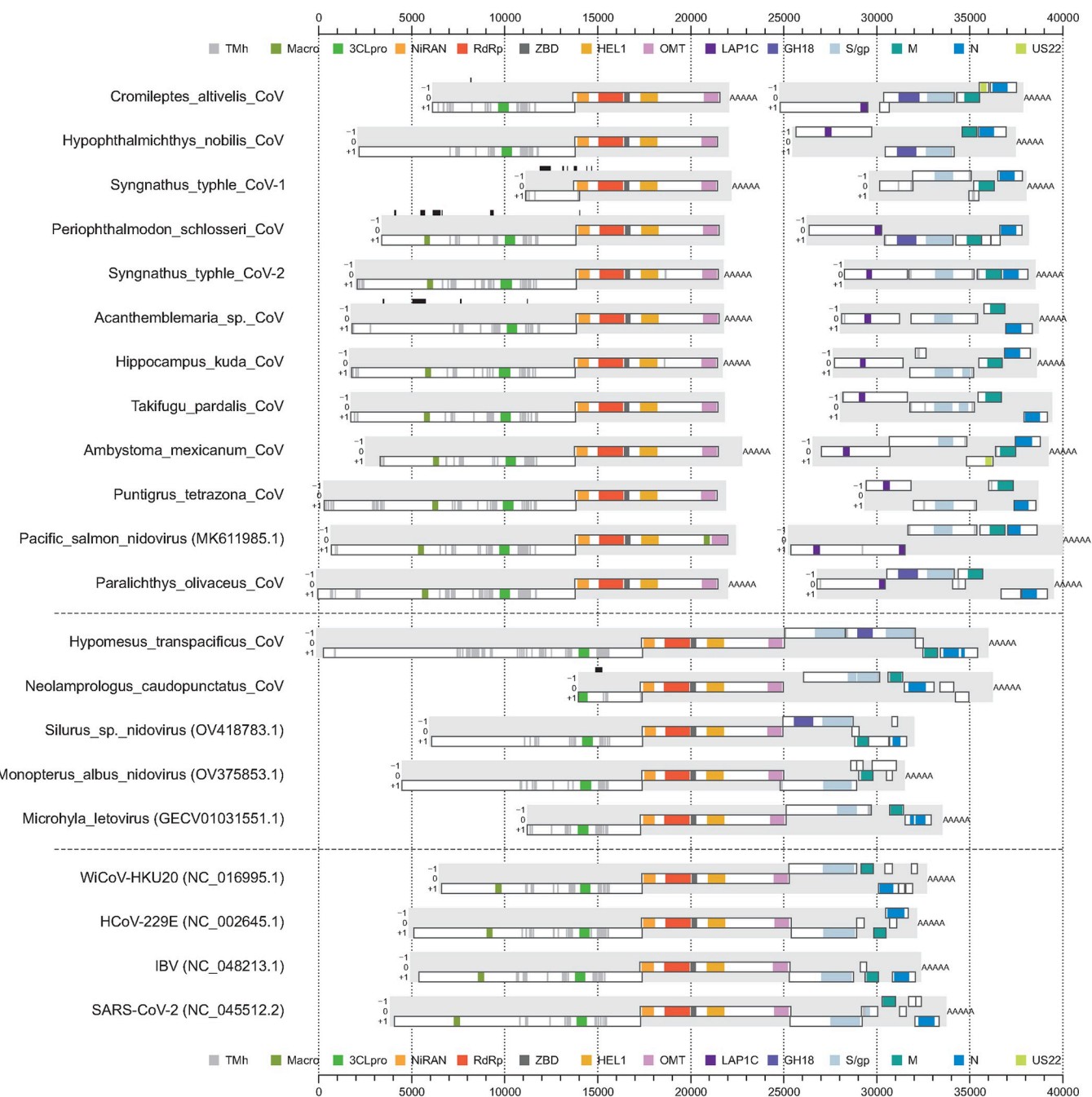

**Fig 4. Genomic layout of novel coronaviruses and five reference viruses.** Viruses that don't start with an accession number in their name are discovered in this study. Predicted open reading frames (ORFs) of at least 300 nucleotides in length are shown as white rectangles; ORFs are defined to start and end at a stop codon. Protein domains predicted via profile HMM are indicated in color; transmembrane helix (TMh), macrodomain (Macro), 3C-like protease (3CLpro), RNA-dependent RNA polymerase (RdRp), RdRp-associated nucleotidyltransferase (NiRAN), zinc-binding domain (ZBD), superfamily 1 helicase (HEL1), O-methyltransferase (OMT), lamina-associated polypeptide 1C-like protein (LAP1C), family 18 glycosidase (GH18), spike/glycoprotein (S/gp), matrix protein (M), nucleocapsid protein (N), US22 protein (US22). Domain borders are drawn according to the corresponding profile search hit and the actual domains may extend beyond these borders. Black bars above a genome indicate missing sequence.

relative to segment 1 (S2 Fig), suggesting a much higher abundance of segments 2 compared to segment 1 with a fold ratio of 8.2 on average (range of 1.3 to 19.2). This difference was statistically significant (Wilcoxon signed-rank test, P = 0.008, n = 8) and may be rationalized as a means to regulate the expression of proteins encoded on segment 2 relative to those on segment 1, similar to the expression via subgenomic RNAs in the monopartite coronaviruses.

Twelve of the nidovirus sOTUs identified in our screen grouped with members of the family *Tobaniviridae* (Figs 3A and S4). Ten of them were found in bony fish sequencing projects, one in a hagfish and a snake sample, respectively. Interestingly, in one of the samples (from *Periophthalmodon schlosseri*) from which we retrieved one of the fish coronaviruses with bisegmented genomes, we also found one of the novel tobani-like viruses, suggesting co-infection of that individual fish with two divergent nidoviruses at the time of sampling.

## Independent validation of bisegmented coronavirus genomes

To independently validate the existence of fish coronaviruses with bisegmented genomes we sequenced and analyzed 202 tissue samples of wild caught individuals of *Syngnathus typhle*. We identified genomic sequences of Syngnathus typhle coronavirus 1 (StyCoV-1) and Syngnathus typhle coronavirus 2 (StyCoV-2), originally discovered in our SRA screen, in, respectively, 22 and 47 of the newly generated sequencing datasets. None of the experiments was positive for both viruses. In all cases, the non-structural and structural proteins were encoded on separate contigs of segment 1 and 2, respectively, supporting genome bisegmentation in these viruses. For both genome segments of each of the two *Syngnathus typhle* coronaviruses we confirmed the presence of poly(A) tails by 3'RACE PCR and Sanger sequencing (Figs 5, S5A and S5B). Moreover, we performed a nested PCR that was designed to bridge the two segments of StyCoV-2 in the orientation that would be expected for an unsegmented coronavirus with canonical ORF1a-ORF1b-3'ORFs genomic organization ('overgap' PCR) using forward primers binding in proximity to the 3' end of segment 1 and reverse primers binding in proximity to the putative 5' end of segment 2. While the corresponding segment-specific control PCRs yielded amplification products of the expected sizes, we did not obtain an amplification product for the 'overgap' PCR, providing additional support for genome bisegmentation (S5C–S5F Fig).

## Emerging genetic markers of bisegmented coronaviruses

In our further dissection of the putative proteome of the newly discovered viruses, we found that the coronaviruses with bisegmented genomes, but none of the other discovered viruses, encode a lamina-associated polypeptide 1C (LAP1C)-like protein. The coronavirus LAP1C-like domain is likely part of a large multidomain protein (protein size from 788 to 2023 amino acids, depending on virus) that otherwise remains unannotated. It is encoded by an apomorphic 5'-proximal ORF on segment 2 upstream of the ORF coding for the spike protein.

Another intriguing observation was that segment 2 of the novel Ambystoma mexicanum coronavirus (AmCoV) from axolotl encodes an additional ORF (between the S and M ORFs) that showed significant sequence similarity with the US22 protein family (HHpred Prob = 94.7%). US22 is known to counter antiviral defense in herpesviruses [60]. Moreover, we found that sequence stretches at or near the 5'-ends of the two AmCoV segments and, similarly, at the 3'-ends of the two AmCoV segments show striking sequence similarity: 97.4% sequence identity across a stretch of 272 nt (positions 431–701 of segment 1 vs. positions 1–272 of segment 2) and 99.8% across 821 nt (positions 19446–20265 of segment 1 vs. positions 12489–13309 of segment 2), respectively. Together, these findings suggest that the two AmCoV segments are packed together in the same virion.

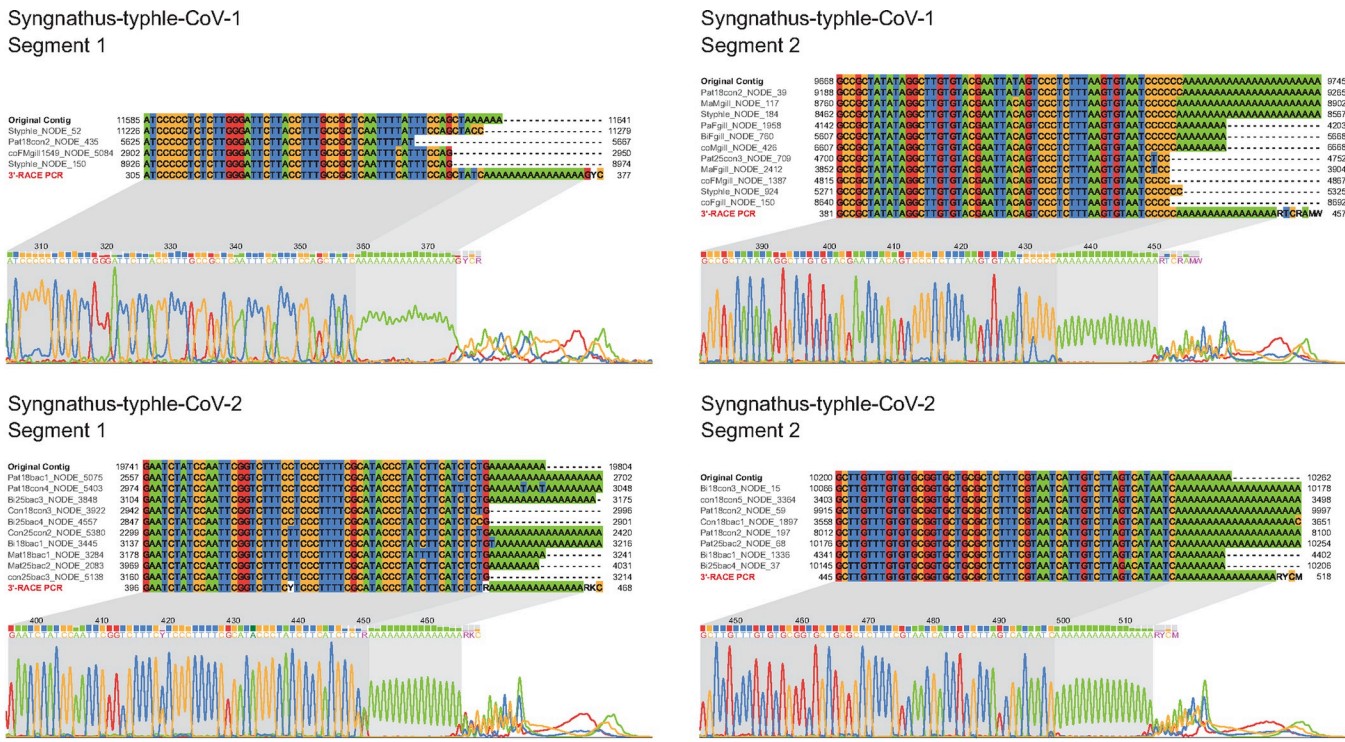

**Fig 5. Molecular validation of the 3'-termini of both segments of two bisegmented fish coronaviruses.** For each segment, a multiple nucleotide sequence alignment of the 3'-ends of the SRA-based contig (Original contig), selected additional strains from different fish specimens and the product of the 3'RACE PCR (red label) is shown. The corresponding Sanger sequencing chromatogram for the 3'RACE PCR is shown below each sequence alignment.

## Dissecting incongruences of corona- and tobanivirus trees for non-structural and structural proteins

Sequence similarities between corona- and tobaniviruses in the S protein are limited to a conserved region in S2. Based on analysis of this similarity in our dataset (see Materials and Methods for details), we delineated three major lineages: one formed by alpha-, beta-, gamma-, deltacoronaviruses, three fish coronaviruses and a tobanivirus (denoted S2G1), a second one uniting 10 tobaniviruses and 15 coronaviruses, the latter including those with bisegmented genomes (S2G2), and a third group formed by members of the subfamilies *Serpentovirinae* and *Remotovirinae* of the family *Tobaniviridae* (S2G3) (Fig 3B). The phylogenetic relationship between these three groups is left unexplored due to limited intergroup sequence similarity of S proteins. We observed complete congruence between the S2 and RdRp+ZBD+HEL1 trees only for viruses of S2G3 (Fig 3), which indicates that no recombination involving the analyzed proteins may have contributed to the divergence of this group of viruses. In contrast, we noticed multiple incongruencies between viruses of S2G1 and especially S2G2 at the subfamily or family level. Thus, we treated S2G3 viruses as a de facto negative control for further analysis of recombination of other viruses.

To clarify a possible contribution of recombination to the observed incongruencies, we analyzed the pairwise evolutionary distance (PED) of each corona-and tobanivirus to its most similar virus in the S2 trees relative to the PED of the same virus pair in the RdRp+ZBD+HEL1 tree via PED ratios, denoted PEDr1 (S3 Table and M&M). We reasoned that upon a model of continuous evolution by substitution and due to stronger conservation of RdRp+ZBD+HEL1 compared to S proteins, the former must have accumulated relatively fewer mutations upon

divergence of a virus pair from a common ancestor (PEDr1 < 1.0). Indeed, this expectation is satisfied for the S2G3 viruses representing a negative control (PEDr1 in the range of 0.45 to 0.70). However, if a pair of analyzed viruses has diverged more profoundly in the RdRp+ZBD +HEL1 region than in S2 (PEDr1 > 1.0), this is most compatible with S2 being acquired through recombination. To identify which virus of a pair or both viruses may have an S2 of recombinant origin, the PED ratio analysis was extended for each virus to the second most similar virus in the S2 tree, denoted PEDr2 and interpreted in the same way as PEDr1 (S3 Table and M&M).

Altogether, we identified five viruses that show considerably larger PED for the RdRp+ZBD +HEL1 region compared to S2 (PEDr1 and PEDr2 in the range of 2.26 to 4.26 and 2.28 to 3.36, respectively; S3 Table and Fig 3). These viruses included Misgurnus anguillicaudatus tobanivirus and four viruses of different (putative) subfamilies of the family *Coronaviridae* (HTCV, Silurus sp. Nidovirus, Microhyla letovirus and Ambystoma mexicanum coronavirus). Three and two of these viruses have, respectively, fishes and amphibians as (predicted) host and all of them cluster in the S phylogeny to other corona- or tobaniviruses which also were isolated from aquatic hosts (30 in total) but not from terrestrial hosts (14 in total) (Fig 3). The viruses with inferred recombinant S2 include both single- and bi-segmented coronaviruses. The S2 of the four identified coronaviruses cluster with that of tobaniviruses of two different S2G2 sub-groups whose RdRp+ZBD+HEL1 and S2 trees are congruent (Fig 3). We detail these viruses in the three paragraphs below.

For two of these viruses, HTCV and Silurus sp. nidovirus, the most similar tobaniviruses are Eptatretus burgeri tobanivirus and Fathead minnow nidovirus, respectively; their PEDr1 values are larger 2.0 as well. Three other tobaniviruses separate these two pairs in the S2G2 subgroup, and the analysis of PEDr2 values identified HTCV and Silurus sp. nido-virus but not Eptatretus burgeri tobanivirus and Fathead minnow nidovirus as having S2 of plausible recombinant origin (S3 Table). For HTCV, this involved one of the two spike-like proteins that is encoded by ORF3 (HTCV p3). In contrast, another spike-like protein of HTCV is encoded by ORF2 (HTCV p2) immediately downstream of ORF1b and its S2G1 lineage position is congruent with the position of HTCV in the RdRp+ZBD+HEL1 tree (Fig 3).

Microhyla letovirus and Ambystoma mexicanum coronavirus belong to yet another S2G2-based subgroup that includes five other viruses with congruent phylogenies (Fig 3). Despite being sister to each other in S2G2 tree, Microhyla letovirus and Ambystoma mexica-num coronavirus have Tachysurus fulvidraco nidovirus and Tachysurus vachellii nidovirus as the most and second-most similar viruses; these viruses were used in PEDr1 and PEDr2 analy-ses, respectively, to support a recombinant S2 origin in Microhyla letovirus and Ambystoma mexicanum coronavirus (S3 Table).

Misgurnus anguillicaudatus tobanivirus has an S2 that belongs to the S2G1 subgroup, which otherwise includes homologs encoded by coronaviruses (Fig 3). The S2 of this virus is most similar to that of Monopterus albus nidovirus, while Neolamprologus caudopunctatus coronavirus is the second most similar virus to the pair. Only Misgurnus anguillicaudatus tobanivirus but not Monopterus albus nidovirus showed an anomalously high RdRp+ZBD +HEL1 divergence relative to S2, according to both PEDr1 and PEDr2 analyses (4.09 and 2.28 versus 4.09 and 0.78 in S3 Table).

## S2 is associated with a chitinase-like domain in a subset of S2G2

During comparative sequence analysis of S proteins, we also found that six of the coronavi-ruses and five of the 11 tobaniviruses, which form one of three monophyletic subclusters of

S2G2, encode a divergent chitinase-like domain (HHpred Prob = 100%, sequence identity = 16%) in the N-terminal part of the S ORF upstream of the region homologous to the coronavirus S2 (Figs 4 and S4). We did not detect a chitinase-like domain in the other tobani- and coronaviruses (Fig 3B), indicating a lineage-specific linkage of this protein domain. The viral chitinase-like domain belongs to glycosidase family 18 (GH18) and forms a separate monophyletic lineage in the GH18 phylogeny (S6 Fig). GH18 was found associated with S2 of only two of five vertebrate nidoviruses that may have acquired S2 through recombination (Fig 3).

## Other novel vertebrate nidoviruses

Besides the 25 corona- and tobaniviruses, we discovered 15 additional vertebrate sOTUs from three other nidovirus families (Figs 3A and S4). This included an arterivirus, detected in an *Eospalax fontanierii* (Chinese zokor) sample that likely forms a new virus species (and a putative new genus), showing only 47% local protein sequence identity in the RdRp to the closest arterivirus reference (GenBank accession QIM73767). In addition, we identified 14 viruses that clustered basal to the *Arteriviridae* within the suborder *Arnidovirineae*. Two of the novel viruses, forming a putatively novel virus subfamily, are from two different turtle species and cluster with Trionyx sinensis hemorrhagic syndrome virus in the family *Cremegaviridae*. Another two novel viruses constitute two putatively novel subfamilies within the family *Nanghoshaviridae*. The remaining 10 discovered viruses, six of which form putatively novel virus genera, are from fish, amphibian and snake sequencing experiments and belong to the family *Nanhypoviridae* (Figs 3A and S4).

## Inference of subgenomic RNA sequences in newly discovered vertebrate nidoviruses

To show that the discovered viral genomes are functional, we collected evidence for subgenomic mRNAs that are used to express the structural and accessory proteins encoded downstream of ORF1b. To this end, we searched for sequencing reads spanning putative leader-body-junctions typical for nidovirus 5'/ 3'- coterminal subgenomic RNAs and analyzed changes in sequencing read depth across the viral genome for two novel viruses with unsegmented genomes—Crotalus viridis tobanivirus (Fig 6A–6D) and Eospalax fontanierii baileyi arterivirus (Fig 6E–6I). In line with the accumulated knowledge for experimentally characterized nidoviruses, we observed sequencing read depth to be increasing steeply and step-wise towards the 3'-end of the viral genome (Fig 6A and 6E) and identified body transcription regulatory sequences (body TRSs) upstream and in proximity to the start codon of each structural/ accessory protein (Fig 6C and 6H) for both viruses. This allowed us to infer five subgenomic RNA sequences for Crotalus viridis tobanivirus (Fig 6D) and ten for Eospalax fontanierii baileyi arterivirus (Fig 6I), each supported by multiple reads spanning the leader-body-junction (Fig 6D and 6I). For the M and N protein of Eospalax fontanierii baileyi arterivirus we identified two equally likely body TRS and thus subgenomic RNA sequences (Fig 6H and 6I).

## Discussion

Our study of the natural virus diversity in more than 260,000 published genomic datasets of eukaryotes using state-of-the-art bioinformatics pipelines expands the scale of the known extraordinary plasticity in nidoviral genome architecture. Its notable functional implications redefine fundamentals of genome expression, virus particle biology, host range and ecology of vertebrate nidoviruses.

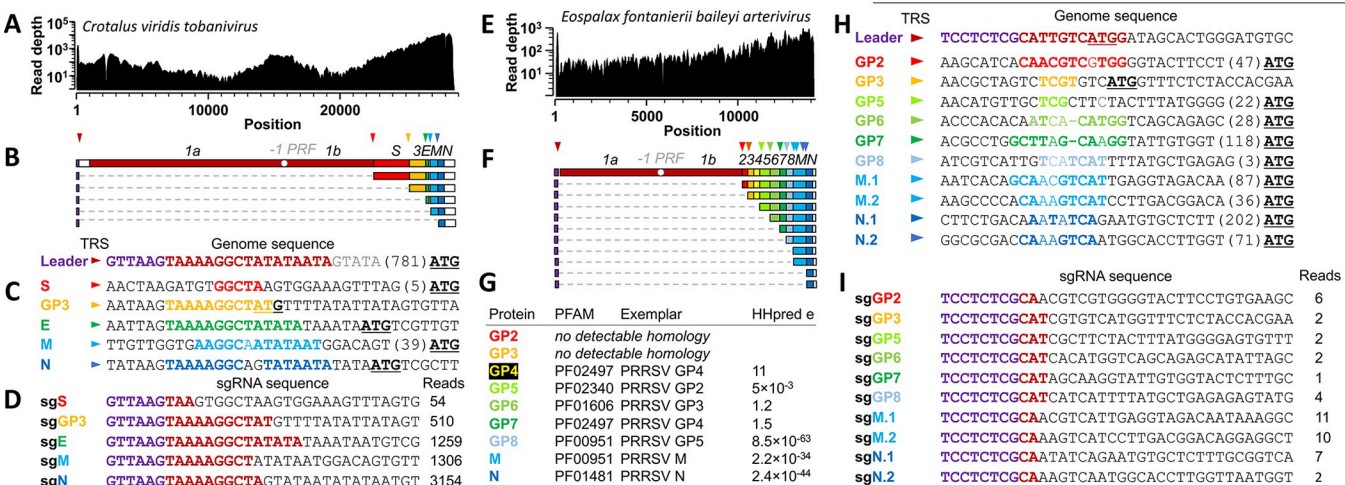

**Fig 6. Sequence-based evidence for subgenomic RNA (sgRNA) formation in Crotalus viridis tobanivirus (A-D) and Eospalax fontanierii baileyi arterivirus (E-I).** Read depth from SRR7401987 (A) and SRR3036364 (E) across the reconstructed virus genomes. (B,F) Inferred reconstruction of viral sgRNAs based on leader-body-junction reads, with the positions of putative transcription regulatory sequences (TRSs) indicated with triangles in the same color as the nearest downstream gene; in cases where multiple body TRSs are used, multiple RNA species are shown. (C,H) Inferred TRSs are shown in colors corresponding to the nearest downstream gene, including distance to the start codon. (D,I) Sequence and read count of sgRNAs showing leader-body fusion; leader sequences are shown in purple, sequences matching the leader TRS in maroon, and sequences from the sgRNA body region in black. (G) Homologs of Eospalax fontanierii baileyi arterivirus structural proteins, inferred from HHpred search against the PFAM-A_v35 database. The best statistical match for each protein and corresponding E-values (HHpred e) are shown.

## The recovered viral genome sequences are reliable and informative

This study builds on advances in sequencing technology, data management and sharing as well as prior bioinformatics-based research that has guided characterization of the life cycle of nidoviruses and nidovirus-host interactions. The use of raw genomic datasets of numerous organisms that were independently compiled by different teams for unrelated projects minimizes biases and serendipity in the reported findings. We observed very few single nucleotide polymorphisms in the assembled sequences, indicating the lack of mixed infection of closely related but different viruses in the analyzed samples. Furthermore, almost identical genome sequences were recovered from several independently collected samples infected with StyCoV-1 and StyCoV-2. We conclude that we determined genome consensus sequences of respective newly discovered viruses. These sequences fit patterns of protein and nucleotide variation and conservation that were established in prior research of coronaviruses or, if they deviated, remained compatible with fundamentals of RNA virus evolution. Accordingly, the newly assembled genome sequences are as reliable as any other that have been obtained by conventional genomic characterization of viruses and which have informed virus research for several decades.

## Coronaviruses with bisegmented genomes

Until few years ago, genome segmentation in positive-sense RNA viruses infecting animals was considered a rare exception. Discoveries of segmented genomes of flaviviruses [61] and even novel, deeply divergent RNA virus lineages [62] have challenged this view. There was also a report of a new lineage of nidoviruses with putatively bisegmented genomes that form a sister lineage to the subfamily *Orthocoronavirinae*, although the authors did not provide molecular evidence for the genome segmentation [53,63]. Here, we describe 12 viruses with bisegmented genomes that belong to 11 genus-like OTUs of the subfamily *Pitovirinae*, family *Coronaviridae*. They include Pacific salmon nidovirus discovered by others [47], which has a genome

(accession MK611985) that is incorrectly annotated as monopartite and shows an assembly gap between the part encoding ORF1a/b (segment 1 according to this study) and the remainder of the genome (segment 2). Our phylogenetic analysis shows that genome bisegmentation is a rare evolutionary event in the *Coronaviridae* confined to a single subfamily in the known vertebrate nidoviruses that also include 20 subfamilies or subfamily-like OTUs comprising viruses with non-segmented genomes (Fig 3A). Association of the subfamily *Pitovirinae* with aquatic hosts is notable, although not exceptional and also observed for non-segmented nidoviruses of ten other subfamilies.

Similar to the segmented flavi-like viruses, which share a common evolutionary history with unsegmented flaviviruses [61] just as the bisegmented coronaviruses do with the conventional coronaviruses with unsegmented genomes, we observed cross-matching nucleotide stretches near the 5′ and 3′ segment termini of AmCoV. They may play a common and essential role in segment packaging into the same virion, like it was documented for genome segments of influenza viruses [64,65]. Moreover, many of the genomic segments assembled by us have a 3'-poly(A) tail. Although our molecular validation by RACE-PCR was done only for the 3'-ends of both genome segments in two of the discovered coronaviruses, we note that characterization of the exact chemical structure of the 5'-end is more challenging and has been achieved only for very few nidoviruses in previous dedicated studies. Together, these results, combined with phylogenetic clustering and large host and divergence ranges of the 12 bisegmented viruses, make it very likely that the assembled segmented genomic sequences are genuine.

The observed recurrent excess in sequence read coverage of segment 2 over segment 1 in *all* bisegmented coronavirus samples, with an estimated ~8:1 ratio on average, must reflect natural differences in the abundance of these segments. This pattern resembles the read coverage excess over the functionally equivalent region observed upon genome sequencing of canonical non-segmented nidoviruses (S2 and S3 Fig), which is due to 3'-region-specific abundance of sg mRNAs. The latter direct a relatively large production of structural compared to non-structural proteins. Thus, genome segment 2 may be overproduced compared to segment 1 in infected cells to regulate expression of the relative amounts of proteins encoded on the two segments in a way that resembles the utilization of genomic and subgenomic RNAs by non-segmented nidoviruses, including newly discovered ones in this study (Fig 6). Furthermore, a step-wise increase of read coverage towards the 3'-end observed for most segment 2 sequences (S2 Fig) indicates that the bisegmented coronaviruses also may employ sg mRNAs for synthesis of some segment 2 proteins. It will be interesting to study how virus particle packaging of equimolar or non-equimolar segments 1 and 2 is regulated in these nidoviruses.

Coronaviruses now offer a promising model system to study the emergence of segmented genomes from unsegmented ones. By using in vitro experiments with coronaviruses and various other (+)RNA viruses, intriguing insights into such major evolutionary transitions were gained [66–70], including the possible emergence of segmented genomes from subgenomic RNAs as proposed for the family *Alphatetraviridae* (formerly *Tetraviridae)* [71]. We hypothesize that coronavirus genome bisegmentation might have taken place in a fish, as all except one virus with bisegmented genome are from bony fish samples, even though the ratio of the total number of analyzed fish and amphibian experiments was 4.6:1. After emergence of genome bisegmentation it possibly followed an inter-class host jump into an amphibian by an ancestor of Ambystoma mexicanum coronavirus that infects axolotls. It is intriguing to see that bisegmented coronaviruses, but not the unsegmented ones that also include fish and amphibian viruses, encode a LAP1C-like coding region upstream of the S ORF, e.g. close to the genomic position where genome fragmentation occurred. The strong linkage between genome bisegmentation and LAP1C presence suggests causation between the two or the involvement of an

unknown factor affecting both. Acquisition of LAP1C might thus have been a trigger on the path towards bisegmentation of the ancestral non-segmented coronavirus genome or vice versa. Interestingly, the same genomic region of segment 2 encodes a second putative S protein in a separate ORF of Hypomesus transpacificus coronavirus, which has by far the largest monopartite genome among all the coronaviruses. These observations indicate that the genomic region between ORF1b and the S ORF can accommodate large genomic size variations. Whether segment reassortment plays a major role in generating genetic variation in the subfamily *Pitovirinae* remains to be explored in future studies.

## Spike gene exchange between corona- and tobaniviruses

Homologous and heterologous recombination are main mechanisms of genetic variation and they generate major genetic novelties. Based on the observed incongruencies of phylogenetic trees for the non-structural and structural proteins (Fig 3) and results of the analysis of PED ratios between replicative and spike proteins (S3 Table), we argue that five viruses (defined as "acceptor") of different subfamilies in the families *Coronaviridae* and *Tobaniviridae* may have acquired the S gene from or exchanged it with a distantly related virus ("donor") of these families. We are not aware of alternative explanations for the observed discordance in phylogenetic signals for RdRp+ZBD+HEL1 and S proteins. Except for Microhyla letovirus prototyping a subfamily, the four other nidoviruses with recombinant S2 solely represent genus-like OTUs in the currently available small sampling. This taxonomic association indicates the scale of divergence of an acceptor virus from its closest known non-recombinant virus; the resolution of this analysis will improve with increasing future virus sampling. A similar taxonomic scaling is not available for donor viruses, since S proteins are not used in taxa demarcation.

The five identified viruses with recombinant S2, which include one tobanivirus and four coronaviruses, present the most striking cases in terms of available evidence. They include HTCV with its 36 kb genome that encodes two spike genes from which one copy (p2) constitutes the original concordant spike protein while the other copy (p3) may have been acquired from a tobanivirus by an HTCV ancestor (Fig 3B). A virus with two different S proteins may also be seen as a functionally competent, evolutionary intermediate to a single S-encoding virus and therefore may provide an alternative two-step mechanism by loss of the original S gene in contrast to homologous gene exchange commonly considered in these cases. Since S proteins form a homotrimeric peplomer in coronavirus virions [72–75], it would be interesting to see if HTCV p2 and p3 form a heteromeric peplomer.

Our analysis of S2-based recombination was complicated by (i) the low support of some internal nodes in the S2 and replicase trees; (ii) the divergence of S2 relative to the replicative proteins that left the relationships between the three S2-based groups unresolved; and (iii) the midpoint-rooting of trees. These factors have limited the resolution of the conducted analysis and may have precluded recovery of most ancient recombination events indicative in other observations of our study. For instance, the 11 GH18-encoding viruses are monophyletic in the S2 tree but have a complex distribution in the RdRp+ZBD+HEL1 tree and involve both mono- and bi-segmented viruses (Fig 3). After accounting for two recombinant viruses, Silurus sp. nidovirus and HTCV p3, the remaining nine viruses belong to two subfamilies of the *Coronaviridae* and *Tobaniviridae*, a pattern that is compatible with ancient recombination. Moreover, the position of human coronavirus 229E (HCoV-229E) in S2G1 next to Wigeon coronavirus HKU20 (WiCoV-HKU20) is incongruent with its position in the RdRp+ZBD +HEL1 tree (Fig 3) and HCoV-299E also shows a PED ratio exceeding the negative control (S3 Table), indicating that the spike of an HCoV-229E ancestor might be derived from an ancient recombination between an alpha- and a deltacoronavirus.

Further analyses, perhaps involving newly discovered nidoviruses in future studies, are therefore required to reveal the full extent of recombinant S proteins between different taxa of vertebrate nidoviruses. Notwithstanding that, our findings suggest that S recombination may be a predominant feature throughput the evolution of nidoviruses infecting aquatic vertebrates. This indicates that specifics of the aquatic hosts, such as their immune system, or the aquatic environment might be conducive for cross-species transmissions of corona- and tobaniviruses.

Notably, two of the viruses of the S phylogenetic lineage formed by corona- and tobaniviruses – Periophthalmodon schlosseri coronavirus and Periophthalmodon schlosseri tobanivirus – were discovered in the same fish specimen. Moreover, we identified nidoviruses from different virus families in the same fish species in two additional cases (*Misgurnus anguillicaudatus* and *Puntigrus tetrazona*), although in different specimens. This observation suggests that co-infection with members from distinct nidovirus families can take place, which is a prerequisite for homologous recombination to happen, and might be fairly frequent. We note that we were unable to include the other structural proteins into the analysis due to their high sequence divergence and therefore could not test whether additional proteins besides GH18 discussed above, or perhaps the full structural module, were affected by or could be a factor in the putative recombination event.

## Novel protein domains encoded in proximity to S protein by corona- and tobaniviruses

Many of the novel nidoviruses discovered in this study encode protein domains that have not been observed in nidoviruses before. Some of these protein domains are encoded in proximity to the S protein, indicating a potential functional link. They include new members of GH18, a family of widespread Glycosidases (EC 3.2.1) found in all three domains of life as well as in viruses. The GH18 family of glycoside hydrosylases comprises chitinases, lysozyme and several others enzymes (see www.cazy.org for further details) with the best hits in our sequence homology searches being chitinases. Although the nidoviral glycosidase shows two conserved aspartates and one conserved glutamate (motif DXDXE) at the expected catalytic sites [76,77], it is subject to future experiments whether it has chitinase or other glycosidase activity. The position of the viral glycosidases as a separate monophyletic lineage within the host glycosidase tree with no close cellular homologs indicates that the viral domain might have diversified into an enzyme with a novel substrate specificity. It could for instance be involved in the release of virions from the host cell, similar to the role of the influenza A virus neuraminidase, a family 34 glycosidase [78,79]. Functional similarity of this GH18 domain with viral hemagglutinin-esterases that are involved in cell entry in some corona- and tobaniviruses [80] is another possible venue for direct experimental testing.

Another novel protein encoded by bisegmented coronaviruses is the LAP1C-like domain. Cellular LAP1C is an integral membrane protein interacting with torsin 1A, an AAA-ATPase [81]. Acquisition of this LAP1C-like protein might be related to the emergence of genome bisegmentation or with adaptation of the ancestral bisegmented viral genome once it emerged. Interestingly, torsin is a mediator of envelopment of host ribonucleoprotein complexes [82], suggesting a possible role of the viral LAP1C-like domain in virion envelopment, for instance via recruitment of torsins.

The presence of the US22-like protein encoded on segment 2 of Ambystoma mexicanum coronavirus is noteworthy. US22 belongs to the SUKH superfamily that comprises diverse proteins employed by a wide range of organisms, from animals to bacteria [60,83]. Members of US22 have been identified in the genomes of herpesviruses and other DNA viruses. In

herpesviruses, US22 has been implicated in counteracting anti-viral responses through interaction with host proteins [83], suggesting a similar role for the coronaviral US22-like domain.

We note that the possible function(s) of many proteins encoded by nidoviruses discovered here remain uncertain due to the lack of detectable sequence similarity to characterized proteins. Future studies should aim at further improving the sampling of the nidovirus genetic diversity, particularly regarding lineages currently represented by only a single viral sequence, as well as at advancing computational tools for functional prediction of highly divergent proteins to ultimately fill this knowledge gap.

## The SRA is a rich source of complete genomes of novel viruses

The potential of the SRA and similar data repositories as a source of data-driven virus discovery approaches is increasingly recognized by the scientific community [33,52–55,84]. An important caveat of such studies, some of which utilize cloud computing infrastructure and are therefore not free of charge [53], is the fact that many of the reported viral sequences are rather short, suggesting that they represent incomplete genome fragments. However, reconstruction of complete viral genomes will be required for filtering out remaining false-positive hits and for a comprehensive description of the viral diversity. Especially the latter point is very critical because of the exchange of genetic material between viral lineages, for instance involving replicative and structural gene modules [32]. Coding-complete genome sequences are also typically a requirement for classification of novel viruses into taxa at the available taxonomic ranks by the ICTV. We addressed this demand for complete genome sequences by designing a dedicated computational pipeline which was essential for revealing numerous insights discussed above.

The SRA and comparable resources offer the largest available, continuously updated and unbiased entry into the hidden viral diversity that exists on our planet [84]. Compared to conventional virus discovery studies that typically involve sample collection and processing, a much larger amount of data is available for analysis with the SRA-based approach. When combined with phylogenetic analysis, a large scale of SRA-based analysis provides a powerful platform that validates new genetic patterns repeatedly observed in newly discovered viruses, like genome bisegmentation or S protein exchange. The SRA also offers high-quality metadata making it frequently possible to link a discovered virus to an organ or tissue or even a host physiological condition. We anticipate that the vast number of novel viruses discovered by the SRA-based approach, when incorporated into the RNA virus phylogeny, will enable us to confidently associate many of these new viruses with their respective host classes. With sufficiently large sample size, we expect that this proper host association may be extended to viruses which otherwise might have been misassigned to a wrong host due to sample contamination. For instance, if one or few viruses discovered from plant samples cluster within a much larger group of viruses found in animal samples it is highly probable that all these viruses infect animals. An insightful example in this respect is presented by at least two of the tobaniviruses that Shi et al. [37] have discovered from invertebrate samples, for which it seems more likely that they infect vertebrates when accounting for phylogenetic relationships of a much wider variety of nidoviruses; one of them, Xinzhou toro-like virus, is included in Fig 3.

## Quality standards for SRA-based viral sequence assemblies

The vast majority of SRA experiments originate from studies unrelated to virus research. Consequently, no enrichment or amplification of viral sequences was pursued, often resulting in low amounts of viral sequences and an excess of non-viral sequences in the dataset. To mitigate this problem, we employed a meta-assembly approach by pooling sequencing experiments

that contained the same virus species. In doing so, we may not fully capture the natural micro-variation of virus populations, but we note that it is of no relevance for identifying novel virus species and has no bearing on the goals and conclusions of this study. In addition, we introduced two metrics that enable a ranking of the viral contigs with respect to assembly quality. Because this approach involves a reference set of established viral genome sequences, which we expect to gradually grow through a continuous flow of new viral sequences from subsequent SRA mining studies, we expect the ranking to be further refined in the future. As starting point, here we used RNA virus genome sequences from two large-scale virus discovery studies and encourage the community to apply and perhaps further advance the proposed assembly quality metrics in future SRA-based virus discovery studies.

For several of the discovered nidoviruses we were only able to retrieve genome fragments from the SRA data, mostly due to insufficient read coverage. Notwithstanding their incompleteness and the associated caveats mentioned in the previous section, we emphasize that these sequences should not be ignored as they provide valuable information by tagging unknown viral diversity at various scales of divergence. Indeed, these viral genome fragments might be considered as important for approaching a comprehensive description of the virosphere as were expressed sequence tags (ESTs) for gene discovery prior to the availability of the human genome sequence [85].

## Materials and methods

### Detection of viral sequences in transcriptome data

The SRA data analyzed included in total 269,184 sequencing runs available by the time of analysis from (i) fishes excluding *Danio rerio* (18% of all analyzed experiments), (ii) amphibians (4%), (iii) sauropsidians (7%), and (iv) mammals excluding *Homo sapiens*, *Mus musculus* and *Rattus norvegicus* (71%). SRA data was downloaded using the SRA Toolkit [86]. The SRA datasets were screened for the presence of nidovirus sequences using the hmmsearch program of the HMMer package with a nidovirus NiRAN and RdRp protein profiles as query. Sequencing reads hit with an E-value smaller than 10 were assembled using CAP3 and the resulting contigs and singlets were compared to the non-viral subset of the NCBI reference proteins (nr) database using blastx, and an E-value cut-off of $10^{-4}$ was used to filter out non-viral sequences. The remaining sequences were compared to the NCBI viral genomics database using tblastx and hits with an E-value smaller than 1 were retained.

### Assembly of viral sequences

SRA data were downloaded using the SRA Toolkit [86]. Sequencing adapters and low-quality bases were trimmed using fastp [87]. A targeted assembly of viral sequences was done using a iterative, seed-based approach as implemented in Genseed-HMM [88]. A nidovirus RdRp protein profile was used as initial seed in the Genseed-HMM analysis. During iterative contig extension, the ends of the current contig are cut and used as query to search for additional reads that align to the contigs ends. We used 45 nucleotides as the size of these cut contigs ends. The newly found reads are then used to extend the contig if they overlap with one of the contig ends by at least 20 nucleotides and with at least 85% sequence identity. Genseed-HMM was run with three different assemblers—CAP3, Newbler and SPAdes [89–91]. The resulting contigs formed the input for a super-assembly using CAP3. The supercontigs were filtered for possible contamination with host sequences by running a Blastx against the non-viral subset of refseq_protein, running another Blastx against the viral subset of refseq_protein and keeping only the contigs that received better hits in the second comparison. If the viral contigs seemed to represent an incomplete viral genome, the whole sequencing projects were analyzed again

via untargeted de-novo assembly. The untargeted virus assembly approach includes downloading of the unprocessed, raw sequencing data followed by trimming sequencing-adapters and low-quality bases using Trimmomatic v. 0.39 [92]. Subsequently, the remaining reads were mapped against the host's genome, if available, using Bowtie 2 v. 2.3.4.1 and SAMtools v. 1.7 [93,94]. For untargeted de-novo assembly the assemblers MEGAHIT (v. 1.2.9), SPAdes (v. 3.11.1) and CAP3 (version data: 12/21/07) were used [90,91,95]. The resulting assemblies were handled with SeqKit [96]. Finally, assembled sequences were classified as viruses based on BLAST hits in the non-redundant database of the NCBI [97]. Furthermore, sequence alignments were performed with T-coffee version 11.00 [98] and MAFFT v7.310 [99] and visualized with IGV [100]. Sequencing reads included in an assembly were mapped back to the respective contigs using Bowtie2 [93], read coverage was visualized using R [101] and assembly quality was assessed by visual inspection via IGV [100].

## Virus assignment to the order *Nidovirales*

Contigs were translated *in silico* in all six reading frames using getorf from the EMBOSS package [102]. Hmmsearch from the HMMer package [103] with a nidovirus 3CLpro, NiRAN, RdRp, ZBD, and HEL1 profiles as query was used to screen these translated peptide sequences in order to identify candidate nidoviral sequences with significant sequence similarity to known nidoviruses. Hits with an E-value of 10 or smaller were kept. A viral sequence was considered as belonging to the order *Nidovirales* either if it encoded the array of protein domains conserved in nidoviruses and composed of 3CLpro-NiRAN-RdRp-ZBD-HEL1 (in that order) or, in the case of partial sequences lacking some of these domains (but not the RdRp), if it grouped within the diversity of other nidoviruses in the RdRp phylogeny.

## Quality assessment of viral genome assemblies

For each assembled contig or reference sequence we downloaded the SRA dataset(s) used to produce the sequence using the SRA toolkit [86]. After preprocessing the sequencing reads as described above, we mapped the reads to the contig/reference sequence using Bowtie2 [93] and only kept aligning reads and for those only the best (primary) alignment. We computed coverage depth and extracted alignment scores using Samtools [94]. We defined the minimum coverage (mico) of a sequence to be the minimum coverage depth across all its positions excluding the terminal 100 nt at both ends. We defined the mean alignment score (meas) to be the average alignment score of all reads aligning to a position averaged across all positions excluding the terminal 100 nt at both ends.

We calculated mico and meas values for the nidovirus contigs assembled in this study and for a reference set of 2350 RNA virus sequences taken from [37] and [57]. We sorted the mico values of the reference set and partitioned them into 10 equally sized, non-overlapping quantiles (deciles). Each reference decile has two borders corresponding to the minimum and maximum mico value within that decile. We then determined for each nidovirus contig the reference decile to which its mico value belongs (e.g. the mico value is larger than the left decile border and smaller than the right decile border) and defined the MICO of the contig to be the decile number (e.g. 1 to 10). Consequently, MICO values of 1 were assigned to nidovirus contigs with mico values in the lowest 10% of mico values of the reference set, while MICO values of 10 were given to nidovirus contigs in the highest 10% of reference mico values. We computed MEAS from meas values in an analogous way. If a virus was identified in multiple SRA datasets (runs) we used the one that resulted in the highest mean read coverage.

## Computational analysis of bisegmented coronavirus genomes

To provide additional support for bisegmentation of some of the discovered coronavirus genomes we conducted the following analysis. For each coronavirus with putatively bisegmented genome we chose the SRA experiment with the highest viral sequencing depth and mapped all reads simultaneously against both segment sequences. We then looked for read pairs that fulfill the following requirements: (i) The reads of a pair align to different segments in the orientation that would be expected if bisegmentation is due to missing sequence that would join the two segments into a single sequence, (ii) One read of a pair aligns in proximity to the 3'-end of segment 1 (or 5'-end of antisense segment 1) and the other read aligns in proximity to the 5'-end of segment 2 (or 3'-end of antisense segment 2), with proximity defined by the expected insert size (plus some random variance) of a sequencing dataset, (iii) Both reads of a pair show reasonably high alignment scores indicative of alignments with few mismatches/indels. If such read pairs exist it would argue against bisegmentation and rather suggest that a piece of the viral genome between the end of ORF1b and the most 5' structural/accessory ORF was not sequenced due to some unknown reason. We did not identify any read pair for any of the analyzed viruses that fulfill the requirements described above, supporting our finding of genome segmentation of this subset of fish coronaviruses.

Adding to the above, the following points provide further evidence for genome bisegmentation. Firstly, many of the assembled segment sequences have poly(A) tails at their 3'-ends; for several viruses this is true for both segments (see main Fig 4). Secondly, many segment 2 sequences have non-coding regions at their 5'-ends. For some viruses, Ambystoma mexicanum coronavirus is the prime example, the data show that our assembly of both segments is virtually complete with poly(A) tails at both 3'-ends and nearly complete 5'UTRs. Thirdly, sequencing depth is very high for many of the segment sequences, for instance is the mean depth larger 1000 for both segments of Ambystoma mexicanum coronavirus (S3 Fig), and also the two assembly quality metrics (MEAS and MICO) show medium to high values for most of the segment sequences (see S2 Fig).

## 3'RACE PCR and Sanger sequencing analysis of bisegmented coronavirus genomes

Rapid amplification of cDNA ends (RACE) for the 3'-ending of coronavirus segments was performed using the 5'/3' RACE Kit (Roche, product Nr. 3353621001). First-strand cDNA synthesis was performed using 2 µg total RNA of sample 'ID084S17MaFgill' for Syngnathus_typhle_coronavirus_1 and sample 'ID088S18Pat25bac2' for Syngnathus typhle coronavirus 2. The cDNA for the two samples was generated by reverse transcription using oligo(dT)-Anchor primers. PCR amplification (Q5 High-Fidelity DNA Polymerase, Cat. M0491L) was done in a nested approach for each segment. In the first PCR, the outer primer for each segment was paired with the anchor primer with cDNA as input. The PCR products were purified using QIAquick PCR Purification Kit (Qiagen, Cat. No.: 28104) and then used as input for the second PCR using the inner primer for each segment paired with the anchor primer. Specific downstream primer sequences are specified in S5A Fig. The second PCR products were again purified and sequenced by the Sanger dideoxy method. The size distribution for the second PCR products were visualised by agarose gel electrophoresis (S5B Fig).

## Overgap PCR to validate genome bisegmentation

To provide supplemental evidence for bisegmentation of coronavirus genomes, we set up an 'overgap' RT-PCR strategy, assuming the genome would be non-segmented, with primers

binding to downstream (3') segment 1 and upstream (5') segment 2, and respective control PCRs on 3' segment 1 and 5' segment 2 endings. Briefly, first-strand cDNA synthesis was performed using 2 µg total RNA from pooled samples containing Syngnathus_typhle_coronavirus_2 (ID088S18Pat25bac2, ID085S18Pat18con2, ID007S2Pat18con2, ID103S23Bi18con3, ID093S19Bi18bac1) with pooled specific primers binding to 5' region of segment 2 and 3' region of segment 1 (S5C Fig). PCR amplification was done in a nested fashion (S5D Fig) using the same kit and polymerase that were used for the 3'RACE PCR. First, we used the outer primer pairs for the overgap PCR as well as the 5' and 3' control PCRs, with cDNA as input. The outer PCR products were then used as input for their respective inner PCR with inner primer pairs. PCR products from either the outer or inner PCR were visualised by DNA electrophoresis (S5E and S5F Fig).

## Phylogenetic analysis and virus classification analysis

For the phylogenetic analysis of replicase proteins (concatenating RdRp, ZBD and HEL1) we compiled multiple sequence alignments and reconstructed phylogenies for two separate datasets: (i) corona- and tobaniviruses and (ii) other vertebrate nidoviruses (excluding corona- and tobaniviruses). To account for high sequence divergence in the phylogenetic analysis of spike proteins, we applied the following approach. We included only the S2 part of spike for the 40 viruses (22 coronaviruses, 18 tobaniviruses) analyzed. We clustered the 40 S2 sequences using MMSeqs2 easy-cluster with standard parameters [104]. This resulted in ten clusters with 11, 7, 6, 4, 4, 3, 2, 1, 1, and 1 sequences, respectively. For each cluster containing more than two sequences we constructed a multiple sequence alignment using MAFFT with parameters '—maxiterate 1000—genafpair—reorder' [105] and built a profile Hidden Markov Model (pHMM) using hmmbuild of the HMMER3 package [103]. We used these six profiles in a pHMM search against the original 40 S2 sequences and recorded for each sequence the hit with best E-value, excluding hits against a sequence of its own cluster profile. The resulting hit list was ordered by increasing E-value and used to iteratively combine the alignments of the 10 clusters using MUSCLE in profile mode [106]. We stopped combining alignments when all hits were considered. This strategy resulted in three alignments composed of (i) all seven members of the *Serpentovirinae*, (ii) a mixture of one tobanivirus (Misgurnus anguillicaudatus tobanivirus) and seven coronaviruses, the latter including representatives of the alpha-, beta-, gamma- and deltacoronaviruses, and (iii) a mixture of 10 tobaniviruses and 15 coronaviruses. These three alignments were used for separate phylogenetic analyses. Poorly aligned alignment sites were removed prior to tree reconstruction.

For each reconstructed replicase or S2 phylogeny, the best fitting amino acid substitution model was selected using Prottest [107]. This was LG+G4+I for all trees except one of the S2 trees for which VT+G4+I was selected. Maximum likelihood trees were reconstructed using PhyML under the substitution model and other parameters determined by Prottest, and branching support was assessed via 100 non-parametric bootstraps [108]. Trees were visualized using the phytools package in R [109].

The viruses were classified into operational taxonomic units (OTUs) at the family, subfamily and genus level using a pairwise-distance based approach as implemented in DEmARC v1.4 [58]. We obtained and combined pairwise distances from the two replicase trees as described above; inter-tree distances (one member of a pair from one tree and the other member from the other tree) were not considered. Briefly, DEmARC proposes thresholds on pairwise genetic divergence to group similar viruses into clusters whose members show genetic distances that are predominantly smaller than the chosen threshold. Optimal thresholds are found in a data-driven way by minimizing the cost and maximizing the persistence associated

with the clustering imposed by the threshold. The clustering cost is proportional to the number of intra-cluster distances exceeding a threshold and persistence reflects the range of pairwise distances within which the clustering does not change. We used patristic distances extracted from our reconstructed nidovirus phylogeny as input for DEmARC. The resulting classification was compared to the phylogeny of non-structural nidovirus proteins to ensure that all delineated clusters are monophyletic. This prompted a single adjustment of the DEmARC classification at the subfamily rank involving Olivier's shrew virus 1 (NC_035127).

## Recombination analysis between viruses of different (sub)families

To identify viruses with recombinant S2, we (i) analyzed incongruencies between S2 and RdRp+ZBD+HEL1 tree topologies via tanglegrams and (ii) compared pairwise evolutionary distances (PEDs) between the S2 and RdRp+ZBD+HEL1 phylogenies. Regarding the latter, we identified for each virus in a S2 phylogeny its partner with the smallest PED in S2. For the same virus pair, we identified the corresponding PED in the RdRp+ZBD+HEL1 tree. We then calculated the ratio of RdRp+ZBD+HEL1 PED versus S2 PED for each pair and ranked the pairs by decreasing PED ratios. Low PED ratios, corresponding to cases where the PED in the RdRp+ZBD+HEL1 tree is smaller than the PED in the S2 tree, is expected for pairs not involved in recombination as the spike proteins evolve at a higher rate than the RdRp+ZBD+HEL1 proteins. High PED ratios, corresponding to cases where the PED in the RdRp+ZBD+HEL1 region is considerably larger than the PED in the S2 tree, indicate possible cases of recombinant S2 (assuming the analyzed replicative proteins are not affected by recombination). The decision which member of a pair (either one or both) harbors a recombinant S2 was made based on topology incongruencies between replicase and S2 phylogenies. Only striking cases (PED ratio > 2) and topology incongruencies at the subfamily or family level were considered. Potential additional, less supported cases, are discussed in the main text.

## Annotation of viral protein domains

We constructed multiple protein sequence alignments of six nidovirus protein domains that are widely or universally conserved in the order *Nidovirales* - 3CLpro, NiRAN, RdRp, ZBD, HEL1 and OMT–using Muscle [106], followed by manual curation. The alignments were converted to profile Hidden Markov models (pHMMs) which formed the queries in a sequence homology search against the *in silico* translated nidovirus contigs using HMMER3 [103]. It has been shown that pHMM-based homology search methods can fail to detect conserved protein domains in very long polyproteins, such as those encoded by nidoviruses and many other RNA viruses, because key parameters of these tools have been obtained via analyses of host proteins of much shorter lengths [110]. An elegant solution that iteratively partitions a polyprotein into shorter pieces that then either receive hits or go to the next round, named LAMPA, has been devised to address this shortcoming [110]. However, LAMPA did not work with sufficient efficiency for high-throughput analysis in our hands. We therefore took a slightly different approach with a similar rationale: we *in silico* translated the contig sequence into the three forward reading frames using transeq of the EMBOSS package [102]. We then moved a sliding window along the translated polypeptide sequences to partition them into pieces of 1000 aa that overlapped by 500 aa. These polypetide sequence pieces were then queried using the nidovirus pHMMs as described above, and the obtained hits (in particular their left and right borders) were mapped back to the contig coordinates. Moreover, we used the HHblits webserver [111] to annotate additional protein domains that are more divergent than the six key nidovirus enzymes listed above.

Transmembrane helices (TMh) were predicted using TMHMM v2.0c [112]. Prediction of signal peptide cleavage sites was done using SinalP v5.0b [113].

## Computational resources

Most of the computational analyses, in particular the Virushunter and Virusgatherer stages, were conducted on the high-performance computing system Taurus of the University of Technology (TU) Dresden consisting of >1800 computing nodes that provide around 60,000 CPU cores. This allowed us to analyze dozens to hundreds of datasets in parallel at a particular point in time, depending on cluster load. Between 24 to 128 CPUs were used for the analysis of an individual SRA experiment. The full analysis took several months for completion and was done in batches, which were defined by host groups (such as fishes, amphibians, mammals, etc) derived from the SRA metadata.

## Supporting information

**S1 Fig. Contig-specific assembly quality assessment.** The continuous meas and mico values calculated for each novel nidovirus sequence were mapped to deciles of the meas and mico distributions of a reference set consisting of 2350 RNA virus sequences to obtain MEAS (left) and MICO (right) metrics. The numbers next to the MICO symbols indicate the original mico value, e.g. the minimum read coverage observed for the contig across its entire length excluding the terminal 100 nt at both ends.
(PDF)

**S2 Fig. Coverage depth of corona-like virus assemblies.** Mean coverage value is indicated and highlighted by the horizontal dashed line.
(PDF)

**S3 Fig. Coverage depth of tobani-like and arteri-like virus assemblies.** Mean coverage value is indicated and highlighted by the horizontal dashed line.
(PDF)

**S4 Fig. Genomic layout of novel and reference tobani-like, arteri-like, cremega-like, nanhypo-like and nangosha-like viruses.** Names of newly viruses discovered in this study are in black, those of reference viruses in gray. Predicted open reading frames (ORFs) of at least 300 nucleotides in length are shown as white rectangles; ORFs are defined to start and end at a stop codon. Protein domains predicted via profile HMM are indicated in color. See legend of main Fig 4 for further details.
(PDF)

**S5 Fig.** Primers used for 3'RACE PCR (A) and overgap PCR (C), genomic locations of nested primer pairs used for overgap PCR (D) and respective DNA electrophoresis of inner 3'RACE PCR (B), outer overgap PCR (E) or inner overgap PCR (F). Purified PCR products (200 ng) for each segment were loaded and resolved in 1.5% or 1.8% agarose gels.
(PDF)

**S6 Fig. Phylogeny of host and corona- and tobanivirus family 18 glycosidases.** Tips corresponding to nidoviruses are highlighted using red circles; gray circles otherwise. Tip labels start with UniProt accessions in the case of cellular proteins and details about sequences such as host information can be obtained via www.uniprot.org. White and black circles at internal nodes indicate SH-like branching support smaller and larger than 0.8, respectively. The branch lengths are in units of aa substitutions per site; scale bar is shown.
(PDF)

**S1 Table. Sequence Read Archive (SRA) metadata of sequencing projects in which the nidovirus genomes reported in this study were discovered.**
(XLSX)

**S2 Table. DEmARC taxonomic classification results of the nidovirus genomes analyzed in this study.**
(XLSX)

**S3 Table. Results of pairwise evolutionary distance (PED) ratio-based S2 recombination analysis.**
(XLSX)

## Acknowledgments

We thank all colleagues in the scientific community who make their sequencing data publicly accessible. We acknowledge the NCBI for providing an elaborate platform to exchange sequencing data. We thank the Center for Information Services and High-Performance Computing (ZIH) at TU Dresden for generous allocations of computer time. We thank Dmitry V. Samborskiy (MSU) for support with DEmARC and other bioinformatics analysis and Daniel P. Depledge (MHH) for helpful discussions. CL and AEG are members of the European Virus Bioinformatics Center (EVBC).

## Author Contributions

**Conceptualization:** Chris Lauber, Alexander E. Gorbalenya, Stefan Seitz.

**Data curation:** Chris Lauber, Xiaoyu Zhang, Arseny Dubin, Stefan Seitz.

**Formal analysis:** Chris Lauber, Xiaoyu Zhang, Josef Vaas, Franziska Klingler, Pascal Mutz, Benjamin W. Neuman, Alexander E. Gorbalenya.

**Funding acquisition:** Chris Lauber, Thomas Pietschmann, Ralf Bartenschlager, Stefan Seitz.

**Investigation:** Chris Lauber, Xiaoyu Zhang, Benjamin W. Neuman, Alexander E. Gorbalenya, Stefan Seitz.

**Methodology:** Chris Lauber, Xiaoyu Zhang, Benjamin W. Neuman, Alexander E. Gorbalenya, Stefan Seitz.

**Project administration:** Chris Lauber, Thomas Pietschmann, Ralf Bartenschlager, Stefan Seitz.

**Resources:** Thomas Pietschmann, Olivia Roth.

**Software:** Chris Lauber.

**Supervision:** Chris Lauber, Stefan Seitz.

**Validation:** Chris Lauber, Xiaoyu Zhang, Benjamin W. Neuman, Alexander E. Gorbalenya, Stefan Seitz.

**Visualization:** Chris Lauber, Xiaoyu Zhang, Benjamin W. Neuman, Alexander E. Gorbalenya, Stefan Seitz.

**Writing – original draft:** Chris Lauber.

**Writing – review & editing:** Chris Lauber, Xiaoyu Zhang, Pascal Mutz, Thomas Pietschmann, Olivia Roth, Benjamin W. Neuman, Alexander E. Gorbalenya, Ralf Bartenschlager, Stefan Seitz.

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
