## [Editor Report · Decision Letter 0]

6 Sep 2023

Dear Lauber:

Thank you very much for submitting your manuscript "Deep mining of the Sequence Read Archive reveals major genetic innovations in coronaviruses and other nidoviruses of aquatic vertebrates" (PPATHOGENS-D-23-01445) for review by PLOS Pathogens. 

As with all papers reviewed by the journal, your manuscript was reviewed by members of the editorial board and by several independent reviewers. Based on its initial assessment, we believe that is fits the best to PLOS Computational Biology instead of PLOS Pathogens.

While we cannot consider your manuscript further for publication in PLOS Pathogens, we would like to offer you the option to transfer your submission, with reviews, to PLOS Computational Biology https://www.editorialmanager.com/PCOMPBIOL/

If you DO wish to transfer your submission, please click this link:

<DeepLinkData><DeepLinkTypeID>27</DeepLinkTypeID><peopleID>1590974</peopleID><userSecurityID>ef7d0ab1-2b22-4b3c-8f42-409f4b07685d</userSecurityID><documentID>43808</documentID><revision>0</revision><manuscriptNumber>PPATHOGENS-D-23-01445</manuscriptNumber><docSecurityID>cd1923d6-0927-4a20-8671-8f5a0b163715</docSecurityID></DeepLinkData>

If you do NOT wish to transfer your submission, please click this link to decline:

<DeepLinkData><DeepLinkTypeID>28</DeepLinkTypeID><peopleID>1590974</peopleID><userSecurityID>ef7d0ab1-2b22-4b3c-8f42-409f4b07685d</userSecurityID><documentID>43808</documentID><revision>0</revision><manuscriptNumber>PPATHOGENS-D-23-01445</manuscriptNumber><docSecurityID>cd1923d6-0927-4a20-8671-8f5a0b163715</docSecurityID></DeepLinkData>

Please note, all PLOS journals are editorially independent and vary in submission requirements.

Should you choose to transfer, your manuscript files, along with the reviewers' comments and their identities will be transferred automatically, and you will receive a confirmation email within 24 hours. Once transferred, your submission will be returned to you so you can check over your record before completing the submission. You may be asked to provide additional information, such as a response to the reviewers' comments. If you have any questions, please contact the editorial office of PLOS Computational Biology https://www.editorialmanager.com/PCOMPBIOL/

We are sorry that the news is not more positive on this occasion, and we hope you will consider PLOS Pathogens for future submissions. Thank you for your support of PLOS and of open-access publishing.

Sincerely,

Guangxiang Luo

Section Editor

PLOS Pathogens

Kasturi Haldar

Editor-in-Chief

PLOS Pathogens

orcid.org/0000-0001-5065-158X

Michael Malim

Editor-in-Chief

PLOS Pathogens

orcid.org/0000-0002-7699-2064

---

## [Decision Letter · Decision Letter 1]

20 Jan 2024

Dear Prof. Dr. Lauber,

Thank you very much for submitting your manuscript "Deep mining of the Sequence Read Archive reveals major genetic innovations in coronaviruses and other nidoviruses of aquatic vertebrates" for consideration at PLOS Pathogens. As with all papers reviewed by the journal, your manuscript was reviewed by members of the editorial board and by several independent reviewers. Based on the reviews, we are likely to accept this manuscript for publication, providing that you modify the manuscript according to the review recommendations.

The reviewers appreciated the attention to an important topic and agree that your team have significantly improved the work and throughly addressed all previous critiques.  However, Reviewer 1 highlight that it is still important to further validate the existence of bi-segmented coronavirus genomes. Given that your team had access to available RNA samples positive for these bi-segmented genomes, one potentially straightforward approach would be to design a PCR strategy to assess if the amplification of a contiguous product is feasible. With the proper controls, the absence of an amplifiable contiguous product would provide additional evidence to support the existence of these bi-segmented genomes.

Sincerely,

Daniel Blanco-Melo, Ph.D.

Academic Editor

PLOS Pathogens

Michael Malim

Section Editor

PLOS Pathogens

Michael Malim

Editor-in-Chief

PLOS Pathogens

orcid.org/0000-0002-7699-2064

Reviewer Comments (if any, and for reference):

Reviewer's Responses to Questions

**Part I - Summary**

Reviewer #1: The authors took great efforts to improve the manuscript and addressed the concerns of the reviewers. However, I am still surprised about the findings of bi-segmented coronaviruses. As a simple proof and since the authors now have RNA from infected fish available, did they try to combine segment 1 and 2 using specific primers and nested PCR?

Reviewer #2: The paper by Lauber et al reports tghe discovery of bisegmented NIdoviruses by computational mining of SRA.

This is a biologically interesting and important discovery, andf in addition, the manuscript reports several methodological innovations that will be useful to increase the robustness of new virus discovery by mining single read databases. Furthermore, this revised version includes limited but convincing experimental validation of the bisegmented genome organization of the discovered viruses via the detection of polyA tail on both segments.

**Part II – Major Issues: Key Experiments Required for Acceptance**

Reviewer #1: (No Response)

Reviewer #2: I find no major issues with the manuscript and believe that in this revision, the authors have more than adequately addressed all the concerns and suggestions of the reviewers. Further experiments would be beyond the scope of this work and should be pursued and published separately. The authors' response to the reviewers' comments is extremely thorough.

**Part III – Minor Issues: Editorial and Data Presentation Modifications**

Reviewer #1: (No Response)

Reviewer #2: I see no issues with this manuscript, it is very well, clearly written.

PLOS authors have the option to publish the peer review history of their article (what does this mean?). If published, this will include your full peer review and any attached files.

Reviewer #1: No

Reviewer #2: **Yes: **Eugene V Koonin

Figure Files:

Data Requirements:

Reproducibility:

References:

---

## [Editor Report · Decision Letter 2]

31 Mar 2024

Dear Prof. Dr. Lauber,

We are pleased to inform you that your manuscript 'Deep mining of the Sequence Read Archive reveals major genetic innovations in coronaviruses and other nidoviruses of aquatic vertebrates' has been provisionally accepted for publication in PLOS Pathogens.

Best regards,

Daniel Blanco-Melo, Ph.D.

Academic Editor

PLOS Pathogens

Michael Malim

Section Editor

PLOS Pathogens

Michael Malim

Editor-in-Chief

PLOS Pathogens

orcid.org/0000-0002-7699-2064

---

## [Editor Report · Acceptance letter]

18 Apr 2024

Dear Prof. Dr. Lauber,

We are delighted to inform you that your manuscript, "Deep mining of the Sequence Read Archive reveals major genetic innovations in coronaviruses and other nidoviruses of aquatic vertebrates," has been formally accepted for publication in PLOS Pathogens.

Best regards,

Michael Malim

Editor-in-Chief

PLOS Pathogens

orcid.org/0000-0002-7699-2064